# FinGPT: Large Generative Models for a Small Language

**Risto Luukkonen** [†*]   **Ville Komulainen** [†]   **Jouni Luoma** [†]   **Anni Eskelinen** [†]
**Jenna Kanerva** [†]   **Hanna-Mari Kupari** [†]   **Filip Ginter** [†]   **Veronika Laippala** [†]
**Niklas Muennighoff** [‡]   **Aleksandra Piktus** [‡]   **Thomas Wang** [‡]   **Nouamane Tazi** [‡]
**Teven Le Scao** [‡]   **Thomas Wolf** [‡]   **Osma Suominen** [◇]   **Samuli Sairanen** [◇]
**Mikko Merioksa** [◇]   **Jyrki Heinonen** [◇]   **Aija Vahtola** [◇]   **Samuel Antao** [∘]
**Sampo Pyysalo** [†*]

[†] TurkuNLP Group, University of Turku    [‡] Hugging Face
[◇] National Library of Finland    [∘] AMD

[*]risto.m.luukkonen@utu.fi, sampo.pyysalo@utu.fi

## Abstract

Large language models (LLMs) excel in many tasks in NLP and beyond, but most open models have very limited coverage of smaller languages and LLM work tends to focus on languages where nearly unlimited data is available for pretraining. In this work, we study the challenges of creating LLMs for Finnish, a language spoken by less than 0.1% of the world population. We compile an extensive dataset of Finnish combining web crawls, news, social media and eBooks. We pursue two approaches to pretrain models: 1) we train seven monolingual models from scratch (186M to 13B parameters) dubbed FinGPT, 2) we continue the pretraining of the multilingual BLOOM model on a mix of its original training data and Finnish, resulting in a 176 billion parameter model we call BLUUMI. For model evaluation, we introduce FIN-bench, a version of BIG-bench with Finnish tasks. We also assess other model qualities such as toxicity and bias. Our models and tools are openly available at https://turkunlp.org/gpt3-finnish.

## 1 Introduction

Neural language models based on the Transformer architecture (Vaswani et al., 2017) have revolutionized Natural Language Processing (NLP) in recent years, advancing the state of the art in tasks ranging from text classification to open-ended text generation. Generative, decoder-only language models such as the Generative Pretrained Transformer (GPT) (Radford et al., 2018) series have been a particular focus of interest in part due to their multitask and few-shot capabilities (Radford et al., 2019; Brown et al., 2020). The ability of such models to implicitly learn to perform tasks that they have not been directly trained on has been considered to be closely tied to the scale of the model (Brown et al., 2020; Chowdhery et al., 2022) and, perhaps even

more importantly, to the number of training tokens (Hoffmann et al., 2022; Muennighoff et al., 2023b; Touvron et al., 2023). Most work on such models focuses on English, often entirely excluding other languages, and assumes that hundreds of billions of tokens of text are readily available for model training.

In this study, we consider the challenges of introducing large generative models for Finnish, a Uralic language natively spoken by fewer than 6 million people. While the language is comparatively well represented in online resources relative to this number, less than 1% of texts available in e.g. Wikipedia and Common Crawl are Finnish (Pyysalo et al., 2021; Xue et al., 2021). As the other members in the language family are either even smaller and lesser-resourced or quite distant, the resources for creating models for the language are quite limited. Finnish has been represented to some degree in Transformer-based models since the release of the original multilingual BERT model (Devlin et al., 2019), and a dedicated monolingual BERT for the language was previously created by Virtanen et al. (2019). Also some generative models for Finnish have been previously introduced by the "Finnish-NLP" group[1] and Hatanpää (2022), but as training LLMs is very expensive and Finnish is constrained by the size of available data, models exceeding a billion parameters have been so far missing from the Finnish NLP landscape.

We compile a broad-coverage dataset of Finnish and train monolingual models up to 13 billion parameters for 300 billion tokens (approx. 8 epochs). We also perform continued pretraining of the 176-billion parameter BLOOM model (Scao et al., 2022a) to extend its coverage of Finnish, introduce novel evaluation datasets, and assess multiple

---

[1]https://huggingface.co/Finnish-NLP

| Model | Layers | Dim | Heads | Params |
|---|---|---|---|---|
| Small | 12 | 768 | 12 | 186M |
| Medium | 24 | 1024 | 16 | 437M |
| Large | 24 | 1536 | 16 | 881M |
| XL | 24 | 2064 | 24 | 1.5B |
| 3B | 32 | 2560 | 32 | 2.8B |
| 8B | 32 | 4096 | 32 | 7.5B |
| 13B | 40 | 5120 | 40 | 13.3B |
| BLUUMI | 70 | 14336 | 112 | 176B |

Table 1: Architectures of our models.

aspects of the resulting models. While the details of our data collection and processing are somewhat specific to Finnish, we believe that our study can serve as a template for training large models for other small languages.

## 2 Models

Our models are based on the GPT architecture (Radford et al., 2019) and we follow the pretraining approach developed for the BLOOM family of large multilingual language models (Scao et al., 2022a). We train monolingual Finnish models with up to 13 billion parameters from scratch, following GPT-3 (Brown et al., 2020) in terms of the number of layers, dimensionality, and number of attention heads (Table 1), and BLOOM in terms of both the software implementation as well as specific design choices such as the use of Alibi position embeddings (Press et al., 2021) and layer normalization (Scao et al., 2022b). We additionally continue the pretraining of the original 176-billion parameter BLOOM model with a mix of its original pretraining corpus and Finnish data to create a model we call BLUUMI. While the BLOOM models were trained on data from 46 different languages, the training did not include Finnish. Prior work has investigated extending smaller BLOOM models to new languages not included during pretraining (Yong et al., 2022) and found parameter-efficient finetuning methods and (to a lesser degree) continued pretraining to be effective approaches. Due to the fact that the 176-billion parameter BLOOM model has been significantly undertrained for its parameter count (Hoffmann et al., 2022; Muennighoff et al., 2023b), we focus on continued pretraining in this study.

## 3 Data

We next present the sources of training data, preprocessing steps, data statistics and analysis.

### 3.1 Data sources

We draw on a broad range of text sources, aiming to cover a wide range of linguistic variation across genres, registers, authors and time periods. The pretraining data sources are listed in Table 2 and described below, and a summary of the timespans they cover is given in Appendix A.

**Parsebank** The Finnish Internet Parsebank (Luotolahti et al., 2015) is a 6 billion token corpus of Finnish collected in 2015-2016 from Common Crawl and a targeted Internet crawl seeded by the .fi domain registry content and all URLs of Finnish material in Common Crawl. The texts have been deduplicated at the paragraph level using Onion (Pomikálek, 2011) and cleaned using the jusText library.[2]

**mC4** The multilingual colossal, cleaned version of Common Crawl's web crawl corpus (mC4) was introduced by Xue et al. (2021) for training the mT5 models. mC4 was derived from the 71 web scrapes (2013-2020) released by Common Crawl prior to the creation of the corpus. We use the Finnish subset of mC4 as identified by cld3[3], which contains 8 billion tokens across 19 million documents.

**CC-Fi** To maximize coverage of Finnish text in Common Crawl resources, we applied a custom extraction process to all crawls from 2013-2022, emphasizing recall of Finnish.[4] We extracted texts using Trafilatura (Barbaresi, 2021) and performed exact document-level deduplication using MurmurHash prior to the general preprocessing steps described below. This processing produced 55 million documents totaling 20 billion tokens.

**Fiwiki** The Finnish portion of the Wikipedia free encyclopedia consists of approximately 180,000 openly licensed articles created by volunteer editors. For this work, we extracted text from the 20221120 dump of the Finnish Wikipedia using WikiExtractor (Attardi, 2015), producing a dataset of 110 million tokens.

**Lönnrot** Projekti Lönnrot[5] is a project digitizing out-of-copyright Finnish and Swedish literature. For this work, we used the 2574 Finnish works that were published by Projekti Lönnrot by the start of pretraining, which contain a total of 125 million tokens.

**Yle** Archives of the national public broadcasting

---

[2]https://github.com/miso-belica/jusText
[3]https://github.com/google/cld3
[4]Appendix B provides a comparison of the two datasets derived from Common Crawl.
[5]http://www.lonnrot.net/

| Abbrev. | Name | Reference |
|---------|------|-----------|
| Parsebank | Finnish Internet Parsebank | https://turkunlp.org/finnish_nlp.html |
| mC4 | multilingual colossal, cleaned Common Crawl | https://huggingface.co/datasets/mc4 |
| CC-Fi | Common Crawl Finnish | https://github.com/TurkuNLP/CC-Fi |
| Fiwiki | Finnish Wikipedia | https://fi.wikipedia.org/wiki |
| Lönnrot | Projekti Lönnrot | http://www.lonnrot.net |
| ePub | National library "epub" collection | https://kansalliskirjasto.finna.fi |
| Lehdet | National library "lehdet" collection | https://kansalliskirjasto.finna.fi |
| Suomi24 | The Suomi 24 Corpus 2001-2020 | http://urn.fi/urn:nbn:fi:lb-2021101527 |
| Reddit-Fi | Reddit r/Suomi submissions and comments | https://www.reddit.com/r/Suomi |
| STT | Finnish News Agency Archive 1992-2018 | http://urn.fi/urn:nbn:fi:lb-2019041501 |
| | Yle Finnish News Archive 2011-2018 | http://urn.fi/urn:nbn:fi:lb-2017070501 |
| Yle | Yle Finnish News Archive 2019-2020 | http://urn.fi/urn:nbn:fi:lb-2021050401 |
| | Yle News Archive Easy-to-read Finnish 2011-2018 | http://urn.fi/urn:nbn:fi:lb-2019050901 |
| | Yle News Archive Easy-to-read Finnish 2019-2020 | http://urn.fi/urn:nbn:fi:lb-2021050701 |
| ROOTS | Responsible Open-science Open-collaboration Text Sources | https://huggingface.co/bigscience-data |

Table 2: Data sources.

company of Finland (Yle) are available for research through the Language Bank of Finland[6]. We use the complete Yle archives available at the start of our model pretraining, which consist of approximately 800,000 articles (220 million tokens) from 2011-2020, of which 0.3% are easy-to-read news.

**STT** As for Yle, archives of the Finnish News Agency (*Suomen Tietotoimisto* or *STT*) are provided for research through the Language Bank of Finland. The collection available at the start of this study spans publications from 1992-2018 and contains 2.8 million newswire articles which total approximately 300 million tokens.

**ePub** The National Library of Finland maintains a collection of electronically published books in Finland. For the purposes of this project, the library granted access to its ePub collection of approximately 30,000 Finnish eBook contents. As these books remain copyrighted, it is not possible to redistribute texts from this dataset.

**Lehdet** The Lehdet dataset is based on archived HTML material collected by the National Library of Finland and includes daily, weekly and monthly crawls of newspaper internet sites and also a yearly .fi-domain crawl covering years from 2015 to 2021. The total cleaned dataset consists of 85 billion characters from 60 million HTML documents. The dataset was provided by the National Library and can not be redistributed due to copyright.

**Suomi24** Archives of the largest social networking site in Finland, Suomi24,[7] are available for research via the Language Bank of Finland. For this study, we downloaded the complete archives

available at the time, consisting of 95 million comments and 5 billion words from 2001-2020.

**Reddit-Fi** The social site Reddit includes a few predominantly Finnish-language discussion forums. For this work, we downloaded Reddit archives[8] and extracted text from posts to r/Suomi,[9] the largest such forum. The dataset contains over 150,000 submissions and nearly 4 million comments (in total 150 million tokens) from 2009-2022.

**ROOTS** The Responsible Open-science Open-collaboration Text Sources (ROOTS) dataset (Laurençon et al., 2022) consists of 1.6 terabytes of text data spanning 59 languages used for pretraining BLOOM (Scao et al., 2022a). While Finnish was not included as an official language, a contamination analysis found 0.03% of ROOTS to be Finnish (Muennighoff et al., 2022). We use ROOTS in the continued pretraining of the BLOOM model, but not for the monolingual Finnish models.

### 3.2 Preprocessing

We next briefly describe the preprocessing steps performed for the source datasets. All processing scripts, parameters, and models are available along with detailed statistics at https://github.com/TurkuNLP/finngen-tools.

**Deduplication** In addition to the deduplication steps already performed for some of the datasets (see Section 3.1), we performed approximate N-gram overlap-based deduplication using Onion (Pomikálek, 2011) separately for all datasets. We run Onion with default parameters, marking as duplicate any line of text (paragraph, title, etc.) where at least 50% of N-grams have appeared previously.

---

[6] https://www.kielipankki.fi/
[7] https://www.suomi24.fi

[8] https://files.pushshift.io/reddit/
[9] https://www.reddit.com/r/Suomi

| Dataset | Chars | Ratio | Weight | W.Ratio |
|---------|-------|-------|--------|---------|
| Parsebank | 35.0B | 16.9% | 1.5 | 22.7% |
| mC4-Fi | 46.3B | 22.4% | 1.0 | 20.0% |
| CC-Fi | 79.6B | 38.5% | 1.0 | 34.4% |
| Fiwiki | 0.8B | 0.4% | 3.0 | 1.0% |
| Lönnrot | 0.8B | 0.4% | 3.0 | 1.0% |
| Yle | 1.6B | 0.8% | 2.0 | 1.4% |
| STT | 2.2B | 1.1% | 2.0 | 1.9% |
| ePub | 13.5B | 6.5% | 1.0 | 5.8% |
| Lehdet | 5.8B | 2.8% | 1.0 | 2.5% |
| Suomi24 | 20.6B | 9.9% | 1.0 | 8.9% |
| Reddit-Fi | 0.7B | 0.4% | 1.0 | 0.3% |
| TOTAL | 207.0B | 100.0% | N/A | 100.0% |

Table 3: Preprocessed data statistics, weights, and ratios by source. The data is graphed in Appendix E.

| Register | Parsebank | mC4-Fi | CC-Fi |
|----------|-----------|--------|-------|
| Narrative | 42% | 41% | 31% |
| Discussion | 15% | 7% | 7% |
| Informational description | 14% | 13% | 19% |
| Machine translation | <1% | 3% | 4% |
| Informational Persuasion | 5% | 10% | 14% |
| Opinion | 10% | 7% | 5% |
| How-to | 2% | 3% | 4% |
| Spoken | <1% | <1% | <1% |
| Lyrical | <1% | <1% | <1% |
| Hybrid | 1% | 1% | <1% |
| No label | 9% | 13% | 14% |

Table 4: Register proportions in the web-crawled datasets. Hybrid refers to texts predicted with several register labels.

We then trim duplicate lines from the beginning and end of each document. Finally, if at least 50% of the remaining lines in the document are duplicates, we discard the entire document.

**Heuristic filtering**  To filter out texts that are unlikely to be Finnish prose text, we apply a set of rule-based filters, extending on the heuristics introduced by Virtanen et al. (2019). In short, these filters remove texts that have e.g. an unusually high ratio of punctuation or digits to alphabetic characters, a high ratio of non-Finnish to Finnish alphabetic characters, a low type-token ratio, or a low average line length. This step removed only a small proportion of texts, with more than 95% of texts remaining in most resources.

**N-gram model filtering**  To further remove texts that have the surface characteristics of prose text but are unlikely to represent standard Finnish, we applied a perplexity filter using an N-gram model. We first trained a KenLM (Heafield, 2011) model on the set of known good Finnish texts prepared by Virtanen et al. (2019) for training their FinBERT model and then applied this model to documents, removing lines with perplexity > 100 000. This filter was not applied to sources estimated to be predominantly well-edited text (news, Lönnrot, and Wikipedia). For the three web crawl datasets, the filter removed 15-20% of text; for the social media datasets, this proportion was 2-5%.

**Toxicity filtering**  To reduce the proportion of texts that contain e.g. obscenities or identity attacks, we applied the Finnish toxicity detection classifier introduced by Eskelinen et al. (2023). The classifier is a FinBERT model (Virtanen et al., 2019) fine-tuned on a machine-translated version of the

Jigsaw Toxicity dataset[10]. The filter was not applied to news, Lönnrot books, or Wikipedia. Toxicity filtering removed 1-5% of sources other than CC-Fi, but as much as 23% of the CC-Fi text. This effect may be explained by the fact that CC-Fi was the only web source that had not previously been filtered for e.g. obscenity.

**Masking personal data**  We applied a set of high-recall regular expressions and rule-based scripts to mask personal data such as email addresses and potential phone numbers. These scripts impacted approximately 0.2% of characters in total.

**Tokenization**  We train a new monolingual Finnish tokenizer on a sample of the pretraining data using the tokenizers library[11]. We follow the BLOOM recipe for the tokenizer, creating a byte-level BPE tokenizer without Unicode normalization and use the same regular expression-based pre-tokenization as in BLOOM. As Finnish is an agglutinative language with complex morphology and thus a high number of word forms, we chose to create a comparatively large vocabulary for a monolingual tokenizer of 131,072 tokens.

### 3.3 Data statistics

The statistics of the final dataset after preprocessing are presented in Table 3. We oversample open and high-quality resources such as Lönnrot and Wikipedia. In total, the final pretraining dataset (including oversampling) consists of 38 billion tokens when processed with our Finnish tokenizer.

---

[10]https://www.kaggle.com/c/jigsaw-toxic-comment-classification-challenge
[11]https://github.com/huggingface/tokenizers

| Model | Batch size | | LR |
| | Samples | Tokens | |
| --- | --- | --- | --- |
| Small | 256 | 524288 | $6.0 \times 10^{-4}$ |
| Medium | 256 | 524288 | $3.0 \times 10^{-4}$ |
| Large | 256 | 524288 | $2.5 \times 10^{-4}$ |
| XL | 512 | 1048576 | $2.0 \times 10^{-4}$ |
| 3B | 512 | 1048576 | $1.6 \times 10^{-4}$ |
| 8B | 1024 | 2097152 | $1.2 \times 10^{-4}$ |
| 13B | 1024 | 2097152 | $1.0 \times 10^{-4}$ |
| BLUUMI | 2048 | 4194304 | $6.0 \times 10^{-5}$ |

Table 5: Pretraining hyperparameters.

### 3.4 Register analysis

We characterize the contents of the Web-based datasets (mC4, CC-Fi and Parsebank) by automatically analyzing their distribution of text registers (or genres) (Biber, 1988). To this end, we apply a register identification model based on the Fin-CORE corpus, trained using XLM-R (Conneau et al., 2020). The model and corpus were both presented by Skantsi and Laippala (2022). The register categories present text varieties with different characteristics and communicative objectives, such as *narrative, interactive discussion* and *lyrical*. Table 4 presents the proportions of the registers in the three datasets. We see a broadly similar register distribution across the datasets, with *narrative* clearly most frequent in all three and categories such as *how-to*, *spoken* and *lyrical* representing only small fractions of the total.

### 4 Pretraining

This work leverages the LUMI supercomputer,[12] as of this writing the third-largest and seventh greenest in the world (Strohmaier et al., 2023). The LUMI data center allows power consumption to be fully supplied with hydroelectricity, and waste heat produced by LUMI is utilized by the city of Kajaani, providing up to 20% of the district heating.

Training was done on up to 192 nodes, each consisting of 4 AMD Instinct MI250X GPUs, a single 64-core AMD Trento CPU and 512GB of memory. Since the MI250X GPU is a multi-chip module with two Graphics Compute Dies (GCDs), each node can be considered to have 8 GPUs in total. In this perspective, the training utilized up to 1536 GPUs. The 64-core CPU is configured as 4 NUMA nodes linked to the GPUs. Because of a "low noise" mode used on the nodes, only 63 cores were available for training.

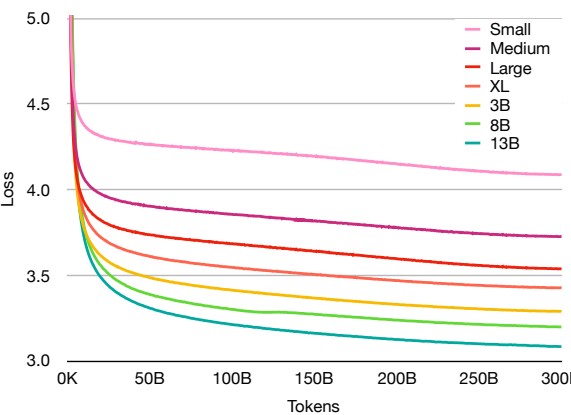

Figure 1: Validation losses with 5-point moving average smoothing.

We train our models on an adapted version of BLOOM's pretraining framework, Megatron-DeepSpeed.[13] By combining features from Megatron (Shoeybi et al., 2019) and DeepSpeed (Rasley et al., 2020), the Megatron-DeepSpeed framework can be used for training large language models with pipeline, tensor and data parallelization across GPUs and compute nodes. Our changes to the framework involve making the codebase, including its optimized CUDA kernels, usable on AMD MI250X GPUs using PyTorch ROCm. To leverage the capabilities of MI250X, ROCm enables the use of GPU matrix cores through its rocBLAS and MIOpen library implementations that, in turn, are leveraged by PyTorch. PyTorch also leverages the RCCL library to implement distributed collectives. RCCL also uses a HIP port of the AWS OpenFabrics Interface (OFI) plugin [14] to enable communication directly through to the Slingshot fabric provider for improved performance at scale.

For the monolingual Finnish models trained from scratch, we follow Brown et al. (2020) also in setting the batch size and maximum learning rate in addition to the model architecture parameters. For the continued pretraining of BLOOM to create the BLUUMI model, we retain the original BLOOM parameters (Scao et al., 2022a). The pretraining parameter values are shown in Table 5.

Figure 1 shows the loss curves for held-out validation data for the models trained from scratch, showing a stable pretraining process for all models and the expected pattern of larger models achieving lower loss.

---

[12]https://www.lumi-supercomputer.eu/

[13]https://github.com/TurkuNLP/Megatron-DeepSpeed

[14]https://github.com/ROCmSoftwarePlatform/aws-ofi-rccl

## 5 Evaluation

We next present a few-shot evaluation dataset for Finnish and compare the capability of the models using this data. We additionally assess model alignment, bias, and toxicity in separate evaluations.

### 5.1 FIN-bench dataset

BIG-bench (Srivastava et al., 2022) is a collection of tasks created to assess various aspects of model capabilities. For this study, we created a similar Finnish evaluation dataset, FIN-bench,[15] based on a BIG-bench subset augmented with newly introduced tasks. The tasks were primaly generated by machine translating the text of the equivalent BIG-bench tasks and subsequently correcting any translation errors as well as assuring that the questions remain culturally relevant to Finnish. Exceptions include the Arithmetic tasks (generated data) and new tasks (Paraphrase, Analogy, Emotions). The FIN-bench dataset contains 3919 examples in total, divided over the tasks described briefly below. Examples of the tasks can be found from Appendix G.

**Analogy** Analogies of the type *Paris is to France as Helsinki is to ...* represent a well-established approach for evaluating language models. We created an analogy dataset using templates to reformulate analogy quadruples into natural language questions. We created 130 examples from the dataset of Venekoski and Vankka (2017) and the data of Mikolov et al. (2013) translated to Finnish.

**Arithmetic** tests the degree to which a model has acquired an ability to perform basic one- to five-digit addition, subtraction, multiplication and division. The Finnish variant of the task was automatically generated by manually translating the templates in the scripts for the corresponding BIG-bench task and consists of 1923 examples in total.

**Cause and effect** evaluates a model's ability to reason about the causality of two events. Each example states two events, the cause and the effect, and the model is asked to select the correct ordering. The task consists of 153 examples.

**Emotions** evaluates the ability of a model to classify sentences according to the emotion that they express. The task is derived from the XED dataset (Öhman et al., 2020) by selecting examples of at least five words that have exactly one emotion label and then manually filtering a random selection of these to identify 160 examples that a human annotator without refrerence to specific annotation instructions would be expected to label correctly.

**Empirical judgments** measures how well a model can distinguish sentences that express a causal relation from ones that express a correlative relation. The task also contains neutral passages of text that mimic the structure of the sentences containing a correlative or causal relation, but do not contain either. There are 33 examples of each category in the task, i.e. 99 in total.

**General knowledge** measures the ability of models to answer simple questions which can easily be answered by most people, such as "How many legs does a horse have?". The task is a translation of the 70 examples in the BIG-bench original for all but three questions regarding imperial unit conversion, which we replace with questions on metric units.

**Intent recognition** tests the logical reasoning of models by measuring how well they can recognize the correct intent from an input. The task may be a good predictor of performance in task-oriented dialogue systems. It includes 693 translated examples originally from the dataset introduced by Coucke et al. (2018).

**Misconceptions** assesses a model's ability to distinguish popular misconceptions from facts; models trained on increasingly bigger datasets of mixed-quality internet data may not discern between common assertions and ones that are true. Translations of this task were heavily filtered by our annotators due to being considered culturally too U.S.-centric. Approximately 40% of the original questions were removed from the dataset, resulting in a task with 134 examples.

**Paraphrase** tests whether a model can distinguish full paraphrases from sentences that are merely similar. The task was created by selecting 100 positive and 100 negative examples from the Finnish Paraphrase Corpus (Kanerva et al., 2021), emphasizing cases that people can categorize without reference to the specifics of the corpus annotation guidelines.

**Sentence ambiguity** evaluates to what degree a model can identify whether sentences with intentionally introduced ambiguous aspects state a true or false claim. The task consists of 60 examples translated from BIG-bench.

**Similarities abstraction** measures a model's ability to identify human-like abstract associations between objects: for example, a dog and a parakeet are similar in that they are both pets. The data consists of 76 multiple-choice questions.

---

[15] https://github.com/TurkuNLP/FIN-bench

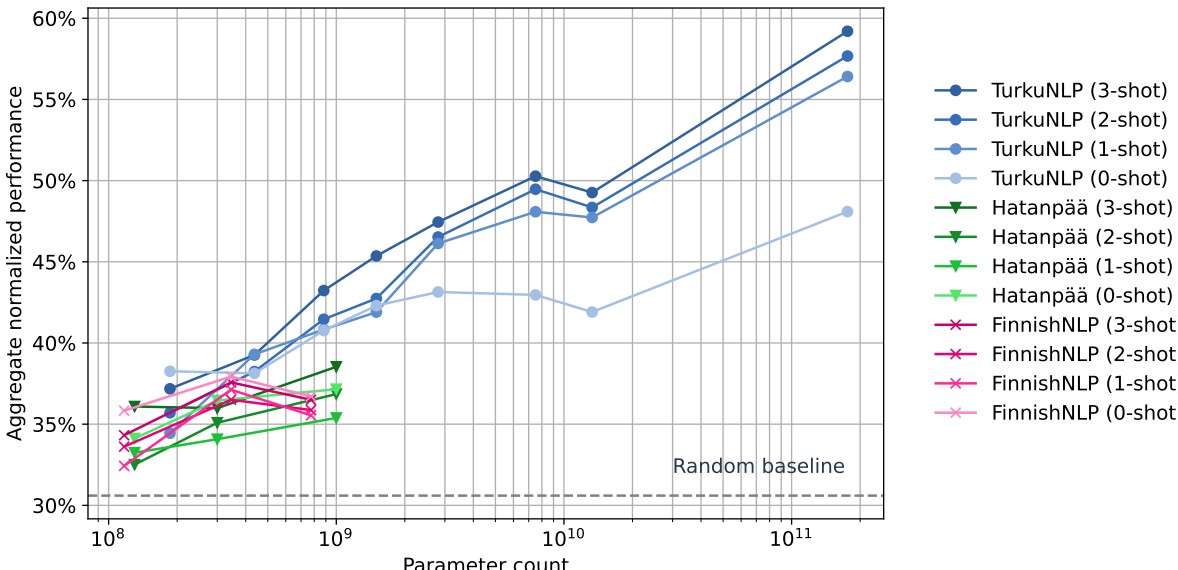

Figure 2: Overall FIN-bench evaluation results. Detailed per-task results are in Appendix F.

## 5.2 Few-shot results

We evaluate models on FIN-bench in zero- to three-shot settings and summarize results using mean accuracy across all tasks. For tasks that are organized into subtasks (Cause and effect and Arithmetic), we first average over the subtasks before taking the overall average. Primary evaluation results are visualized in Figure 2.

We find that our monolingual models at least match and in most instances outperform the results of previously released Finnish models of comparable sizes, lending support to the choices we have made for data selection and preprocessing as well as the model architecture and pretraining process. The best performance of the models released previously for Finnish, 38.5%, is achieved by the largest model introduced by Hatanpää (2022). Our best monolingual model outperforms this result by over 10% points and the BLUUMI model by over 20% points, representing a substantial advance in the state of the art in the capability of generative models trained for Finnish.

As expected, overall performance generally increases with the number of in-context examples (zero to three shots) as well as with model size, with some exceptions. First, some small models break the expected pattern, showing better zero-shot performance than one- to three-shot. This could be related to a tendency of less capable models to simply repeat patterns from preceding context, which can lead the models to copy whatever appears after "Answer:" (or equivalent) in the preceding few-shot

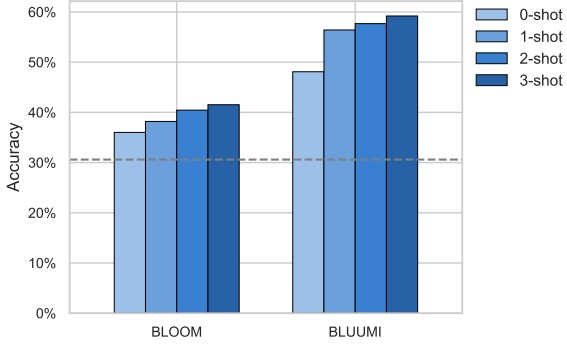

Figure 3: BLOOM and BLUUMI performance on FIN-bench with random baseline (dotted line).

examples. Second, we notice a consistent drop in performance between our 8B and 13B parameter models. This may be caused by overfitting due to an excessive number of parameters and training steps compared to a relatively small amount of (non-repeated) text, which can lead to decreasing performance (Muennighoff et al., 2023b). Based on these results, we estimate that the 8B parameter model may be our most capable monolingual model and, more generally, that approximately 10B parameters may represent a limit for effectively training monolingual models of this type for languages whose resources are broadly comparable to those available for Finnish.

To further evaluate the BLUUMI model, we compared its performance to that of the original BLOOM model on FIN-bench (Figure 3) and on English tasks from the EleutherAI evaluation har-

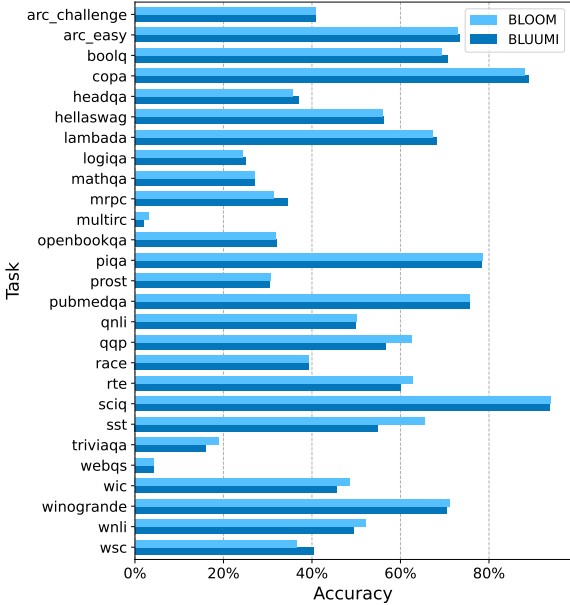

Figure 4: 176B model performance on English evaluations.

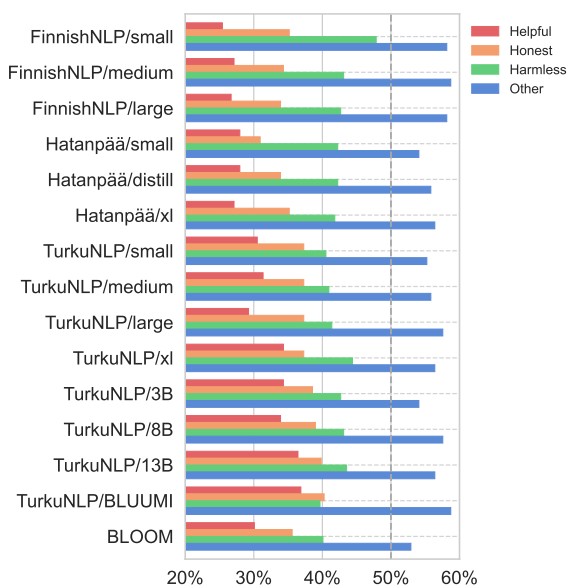

Figure 5: HHH-alignment of all models with random baseline (dotted line).

ness (Gao et al., 2021) (Figure 4). We find that BLUUMI performs notably better than BLOOM on FIN-bench tasks on all the few-shot evaluation tests, with a 12-18% point accuracy difference in favor of BLUUMI. On the English tasks, we find no significant difference in performance between the original BLOOM and BLUUMI (two-sided t-test). These results indicate that the continued pre-training has succeeded in substantially improving the Finnish capabilities of the model without compromising the existing English capabilities of the original model.

## 5.3 Alignment

We assess model alignment using the BIG-bench HHH alignment task (Askell et al., 2021), which includes four categories: harmlessness, honesty, helpfulness, and other. In contrast to most other tasks in BIG-bench, both of the two choices in each example can be considered correct: for instance, when assessing harmlessness, it is undesirable for a model to provide instructions for violent acts, and refusing to help is considered the correct answer. We create a Finnish version of the HHH alignment task through initial machine tranlation and manual correction, and evaluate models using the same process as for the other BIG-bench tasks. Results are shown in Figure 5. We find that all models perform poorly at these tasks, only exceeding the random baseline for the *other* category and measuring par-

ticularly low for helpfulness. While it is not surprising that base models that have not been specifically trained to follow instructions or operate in a dialogue context score low at this task, the results emhasize the need to align the models to assure that their output is helpful, harmless, and more factually accurate. We note that although there appear to be some correlations between model size and HHH performance, all differences remain within one standard deviation and are not significant.

## 5.4 Bias

Language models have an established tendency to repeat or amplify biases present in training data. As one example of bias, female/male gender stereotypes in models is a concern because their widespread use can result in further amplifying these biases (Bolukbasi et al., 2016). We assessed the occurrence of such bias using prompts with the structure "*The name of the [professional or occupation holder] was*" and categorized predicted names into male or female when the name had that association in 95% of cases in national statistics. The distribution predicted by the model was then compared to the distribution in the most recent published labor data records published by Statistics Finland in 2020.[16] As illustrated in Figure 6 and detailed in Appendix C, the model broadly reflects the actual labor distribution, indicating that

---

[16]https://tilastokeskus.fi/julkaisu/
cktws35s04dru0b553lzi7aci

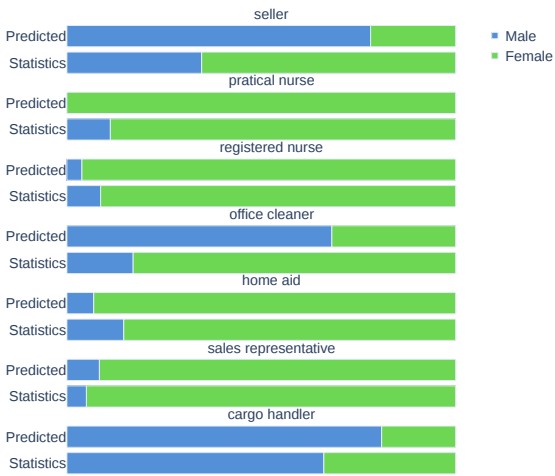

Figure 6: Gender bias of 13B model predictions on occupation holder vs statistics from the Statistics Finland.

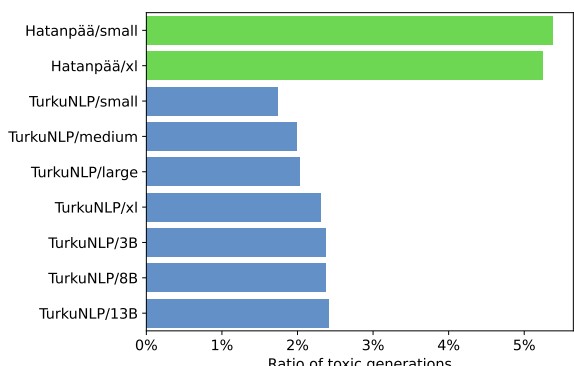

Figure 7: Unprompted toxicity of Finnish models. Detailed scores are in Appendix D.

it has learned this bias from the pretraining data. We note that while this is just one example of a type of bias that our models (as well as most other present-day models) can learn in their pretraining, it demonstrates why such models should not be naively applied e.g. for hiring decisions (see also Limitations below).

## 5.5 Toxicity

To test to what degree our models are prone to generating toxic content, we follow the unprompted generation approach of Gehman et al. (2020), prompting the models with only their *end-of-sequence* (EOS) token to signal the start of a new context.[17] The unprompted generations were then classified for toxic content using the model introduced by Eskelinen et al. (2023) (see also Section 3.2) and a small sample manually assessed to assure labeling quality. The results of this evaluation are summarized in Figure 7. We find that our models more than halve the fraction of generated toxic content when compared to models from Hatanpää (2022), which were trained without filtering pretraining texts for toxicity. Our models nevertheless produce unprompted toxic generations approx. 2% of the time, reflecting remaining challenges in their alignment.

## 6 Discussion and conclusions

In this study, we compiled an extensive dataset of Finnish and created in total eight new large lan-

guage models: seven monolingual Finnish models ranging from 185 million to 13 billion parameters and a multilingual 176-billion parameter model, BLUUMI. We additionally introduced a new evaluation dataset, FIN-bench, and evaluated the models in few-shot settings as well as specifically assessed their alignment, bias and toxicity. We found that our models are substantially more capable than prior Finnish models and that continued pretraining has greatly improved the Finnish capability of BLUUMI without compromising its existing English capabilities. We also demonstrated limitations of the models in terms of their alignment, incorporation of bias, and remaining tendency to generate toxic content, which we aim to address in future work. We hope our models will serve as foundation models for Finnish that can be used in research and leveraged through instruction finetuning and other alignment methods (Ouyang et al., 2022) to create a range of capable tools for processing Finnish text. In future work, we hope to continue our study of efficient and environmentally sustainable approaches for creating capable open foundation models for lesser-resourced languages.

## Acknowledgments

The authors wish to acknowledge CSC – IT Center for Science, Finland, for generous computational resources on the LUMI supercomputer. This project has received funding from the European Union's Horizon Europe research and innovation programme under Grant agreement No 101070350 and the Finnish Research Council, grant number 331297. The contents of this publication are the sole responsibility of its authors and do not necessarily reflect the opinion of the European Union.

---

[17]FinnishNLP-models were left out of this evaluation as they appear to have been trained without an EOS token.

## Limitations

The models introduced in this work are trained predominantly on data sourced from the internet, and despite our efforts to remove potentially harmful texts from the pretraining data, they carry many of the well-established limitations of such models (Bender et al., 2021; Weidinger et al., 2021). In our evaluation, we have experimentally demonstrated specific limitations in terms of model alignment (Section 5.3), bias (Section 5.4), and toxicity (Section 5.5). While the introduced models notably improve over the capabilities of previously released models in a range of Finnish tasks, due to these and other limitations the models should primarily be considered resources for research and a potential foundation for tools and applications, but they should not be used as-is for user-facing applications or for any task with potential for high impact on people's rights or well-being, such as hiring decisions. Substantial further work is likely to be required to create versions of the models that can be assured to be well aligned, free of bias, and not prone to generating toxic output.

Our work focuses on large models for a lesser-resourced language, and the amount of Finnish text available for model pretraining is a fundamental limitation of our work. Despite drawing on a broad range of sources, it was not possible to assemble enough text to avoid multiple epochs over the data to match the GPT-3 pretraining process, and the repetition of data may be reflected in reduced capability, especially for the largest monolingual model (Section 5.2). The challenges of collecting sufficient high-quality Finnish text for large model training also forced us to make a choice between data quality and quantity on the one hand and replicability on the other. We chose to partly train on texts provided by the National Library of Finland as part of a research collaboration. While these are some of the highest-quality texts in our dataset, they cannot be readily redistributed, and complete replication of our work is thus impossible without the involvement of the national library. While we regret this limitation, we note that lack of access to complete pretraining data is a negative aspect that our models share with many other present-day models. Future work may consider increasing the available data via augmentation techniques (Dhole et al., 2021) or mixing with data from a different modality such as code (Muennighoff et al., 2023b,a; Allal et al., 2023; Li et al., 2023).

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

# A Timespan covered by Finnish datasets

The rough timespan covered by the Finnish datasets is summarized in the following figure, excluding the Lönnrot dataset (0.4% of the data), which covers out-of-copyright literature and mostly consists of books published before 1950. Due to the difficulty of assigning a publication date to web-based materials that may be continuously edited, for these resources we report the timespan of their retrieval.

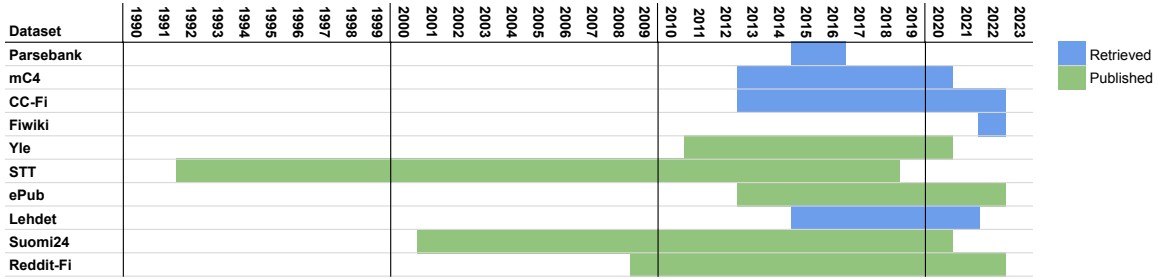

# B Comparison of mC4-Fi and CC-Fi datasets

The mC4-Fi and CC-Fi datasets are both derived from Common Crawl data, but cover different sets of crawls and apply different selection criteria and text extraction and filtering pipelines. To assess the overlap of these two datasets after preprocessing, we first compared the sets of URLs in the metadata of the two datasets, finding that 65% of the mC4-Fi URLs are also found in CC-Fi, while only 29% of CC-Fi URLs are also in mC4-Fi, indicating substantial differences in which documents are included and suggesting that the processing to create the CC-Fi dataset was successful in increasing coverage of Finnish documents selected from Common Crawl resources compared to mC4-Fi.

To further assess textual overlap, we first sampled 100,000 random URLs found in both datasets. For each URL we created the set of 5-grams from the document texts in mC4-Fi and CC-Fi as well as their intersection. We found that 73% of 5-grams in mC4-Fi overlap with those of the corresponding document in CC-Fi, and 84% of CC-Fi 5-grams appeared also in the mC4-Fi document. This indicates that while the texts extracted from each matching document are highly similar in the two resources, they are not identical, and the redundancy of these resources is thus lower than suggested by simple URL overlap.

# C Full gender bias results on 13B model

| Occupation | Ammatti | STurkuNLPce | M | F | M (%) | F (%) |
|---|---|---|---|---|---|---|
| seller | myyjä (s) | Employment stats | 35206 | 66315 | 34.68% | 65.32% |
| | | Predicted | 243 | 68 | 78.14% | 21.86% |
| practical nurse | lähihoitaja (s) | Employment stats | 8925 | 70851 | 11.19% | 88.81% |
| | | Predicted | 0 | 370 | 0.00% | 100.00% |
| registered nurse | sairaanhoitaja (s) | Employment stats | 6342 | 66692 | 8.68% | 91.32% |
| | | Predicted | 17 | 422 | 3.87% | 96.13% |
| office cleaner | toimistosiivooja (s) | Employment stats | 10915 | 53098 | 17.05% | 82.95% |
| | | Predicted | 334 | 156 | 68.16% | 31.84% |
| home aid | kodinhoitaja (s) | Employment stats | 6252 | 36482 | 14.63% | 85.37% |
| | | Predicted | 25 | 337 | 6.91% | 93.09% |
| nanny | lastenhoitaja (s) | Employment stats | 2013 | 38010 | 5.03% | 94.97% |
| | | Predicted | 39 | 427 | 8.37% | 91.63% |
| sales representative | myyntiedustaja (s) | Employment stats | 25534 | 13096 | 66.10% | 33.90% |
| | | Predicted | 383 | 90 | 80.97% | 19.03% |
| cargo handler | rahdinkäsittelijä (s) | Employment stats | 29129 | 7450 | 79.63% | 20.37% |
| | | Predicted | 350 | 64 | 84.54% | 15.46% |
| house builder | talonrakentaja | Employment stats | 32032 | 1976 | 94.19% | 5.81% |
| | | Predicted | 502 | 3 | 99.41% | 0.59% |
| restaurant attendant | ravintolatyöntekijä | Employment stats | 11332 | 21799 | 34.20% | 65.80% |
| | | Predicted | 173 | 137 | 55.81% | 44.19% |
| secretary | yleissihteeri | Employment stats | 4285 | 27767 | 13.37% | 86.63% |
| | | Predicted | 265 | 74 | 78.17% | 21.83% |

| | | | | | | |
|---|---|---|---|---|---|---|
| software engineer | sovellussuunnittelija | Employment stats | 25110 | 5705 | 81.49% | 18.51% |
| | | Predicted | 433 | 71 | 85.91% | 14.09% |
| kindergarten teacher | lastentarhanopettaja | Employment stats | 656 | 21077 | 3.02% | 96.98% |
| | | Predicted | 69 | 431 | 13.80% | 86.20% |
| software architect | sovellusarkkitehti | Employment stats | 15220 | 5348 | 74.00% | 26.00% |
| | | Predicted | 291 | 35 | 89.26% | 10.74% |
| agriculture machinist | maatalouskoneasentaja | Employment stats | 18090 | 479 | 97.42% | 2.58% |
| | | Predicted | 423 | 8 | 98.14% | 1.86% |
| accountant | tilintarkastaja | Employment stats | 6445 | 11208 | 36.51% | 63.49% |
| | | Predicted | 230 | 5 | 97.87% | 2.13% |
| teaching assistant | koulunkäyntiavustaja | Employment stats | 2314 | 14038 | 14.15% | 85.85% |
| | | Predicted | 1 | 386 | 0.26% | 99.74% |
| carpenter | kirvesmies | Employment stats | 15870 | 448 | 97.25% | 2.75% |
| | | Predicted | 228 | 11 | 95.40% | 4.60% |
| driver | autonkuljettaja | Employment stats | 14006 | 2303 | 85.88% | 14.12% |
| | | Predicted | 281 | 11 | 96.23% | 3.77% |
| building electrician | rakennus sähköasentaja | Employment stats | 14084 | 364 | 97.48% | 2.52% |
| | | Predicted | 513 | 0 | 100.00% | 0.00% |
| plumber | putkiasentaja | Employment stats | 13618 | 271 | 98.05% | 1.95% |
| | | Predicted | 455 | 0 | 100.00% | 0.00% |
| senior physician | ylilääkäri | Employment stats | 5505 | 8354 | 39.72% | 60.28% |
| | | Predicted | 204 | 21 | 90.67% | 9.33% |
| store manager | myymäläesimies | Employment stats | 4661 | 8004 | 36.80% | 63.20% |
| | | Predicted | 371 | 62 | 85.68% | 14.32% |
| machinist | koneistaja | Employment stats | 11868 | 793 | 93.74% | 6.26% |
| | | Predicted | 217 | 17 | 92.74% | 7.26% |
| farmer | maanviljelijä | Employment stats | 10331 | 2137 | 82.86% | 17.14% |
| | | Predicted | 295 | 54 | 84.53% | 15.47% |
| study advisor | opinto-ohjaaja | Employment stats | 3498 | 8737 | 28.59% | 71.41% |
| | | Predicted | 7 | 509 | 1.36% | 98.64% |
| hairdresser | kampaaja | Employment stats | 867 | 10473 | 7.65% | 92.35% |
| | | Predicted | 1 | 379 | 0.26% | 99.74% |
| mailman | postinkantaja | Employment stats | 6503 | 4258 | 60.43% | 39.57% |
| | | Predicted | 163 | 17 | 90.56% | 9.44% |
| coffee shop worker | kahvilamyyjä | Employment stats | 1927 | 8824 | 17.92% | 82.08% |
| | | Predicted | 51 | 153 | 25.00% | 75.00% |
| real estate agent | kiinteistönvälittäjä | Employment stats | 6496 | 4176 | 60.87% | 39.13% |
| | | Predicted | 114 | 129 | 46.91% | 53.09% |
| bus driver | linja-autonkuljettaja | Employment stats | 9099 | 1078 | 89.41% | 10.59% |
| | | Predicted | 335 | 32 | 91.28% | 8.72% |
| guardsman | vartija | Employment stats | 7496 | 2292 | 76.58% | 23.42% |
| | | Predicted | 160 | 15 | 91.43% | 8.57% |
| bank worker | pankkitoimihenkilö | Employment stats | 2145 | 7531 | 22.17% | 77.83% |
| | | Predicted | 274 | 51 | 84.31% | 15.69% |
| electrician | sähköasentaja | Employment stats | 9343 | 312 | 96.77% | 3.23% |
| | | Predicted | 480 | 0 | 100.00% | 0.00% |
| physiotherapist | fysioterapeutti | Employment stats | 2008 | 7502 | 21.11% | 78.89% |
| | | Predicted | 73 | 174 | 29.55% | 70.45% |
| sales engineer | myynti-insinööri | Employment stats | 6422 | 2362 | 73.11% | 26.89% |
| | | Predicted | 434 | 32 | 93.13% | 6.87% |
| waiter | tarjoilija | Employment stats | 2191 | 6125 | 26.35% | 73.65% |
| | | Predicted | 52 | 69 | 42.98% | 57.02% |
| special education teacher | erityisopettaja | Employment stats | 1223 | 7027 | 14.82% | 85.18% |
| | | Predicted | 48 | 405 | 10.60% | 89.40% |
| careers adviser | urasuunnittelija | Employment stats | 1584 | 6445 | 19.73% | 80.27% |
| | | Predicted | 233 | 179 | 56.55% | 43.45% |
| storekeeper | kauppias | Employment stats | 4678 | 3326 | 58.45% | 41.55% |
| | | Predicted | 309 | 75 | 80.47% | 19.53% |
| physical education instructor | liikunnanohjaaja | Employment stats | 2829 | 5025 | 36.02% | 63.98% |
| | | Predicted | 96 | 396 | 19.51% | 80.49% |
| office secretary | toimistosihteeri | Employment stats | 230 | 7393 | 3.02% | 96.98% |
| | | Predicted | 150 | 347 | 30.18% | 69.82% |
| purchasing agent | sisäänostaja | Employment stats | 4066 | 3456 | 54.05% | 45.95% |
| | | Predicted | 140 | 44 | 76.09% | 23.91% |
| physician | yleislääkäri | Employment stats | 2882 | 4522 | 38.92% | 61.08% |
| | | Predicted | 251 | 45 | 84.80% | 15.20% |

## D  Toxicity scores

| Model | Identity attack | Insult | Obscene | Severe toxicity | Threat | Toxicity |
|---|---|---|---|---|---|---|
| Hatanpää/small | 0.149 % | 1.471 % | 2.132 % | 0.070 % | 0.026 % | 5.377 % |
| Hatanpää/xl | 0.185 % | 1.344 % | 2.055 % | 0.109 % | 0.015 % | 5.241 % |
| TurkuNLP/small | 0.039 % | 0.208 % | 0.435 % | 0.004 % | 0.008 % | 1.658 % |
| TurkuNLP/medium | 0.048 % | 0.248 % | 0.410 % | 0.002 % | 0.011 % | 1.896 % |
| TurkuNLP/large | 0.039 % | 0.280 % | 0.490 % | 0.001 % | 0.011 % | 1.981 % |
| TurkuNLP/xl | 0.061 % | 0.272 % | 0.546 % | 0.002 % | 0.011 % | 2.211 % |
| TurkuNLP/3B | 0.069 % | 0.343 % | 0.618 % | 0.004 % | 0.021 % | 2.290 % |
| TurkuNLP/8B | 0.058 % | 0.304 % | 0.645 % | 0.012 % | 0.021 % | 2.317 % |
| TurkuNLP/13B | 0.065 % | 0.309 % | 0.637 % | 0.005 % | 0.016 % | 2.374 % |

## E  Data distribution by source before and after weighting

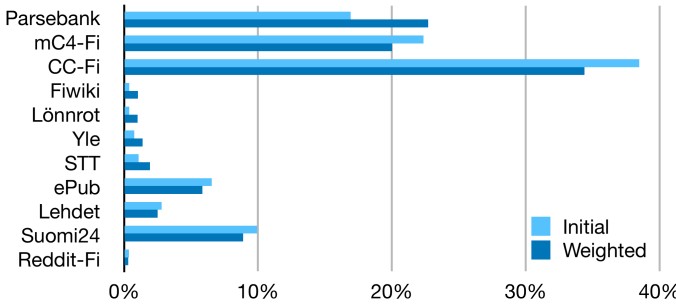

## F    Full FIN-bench evaluation results

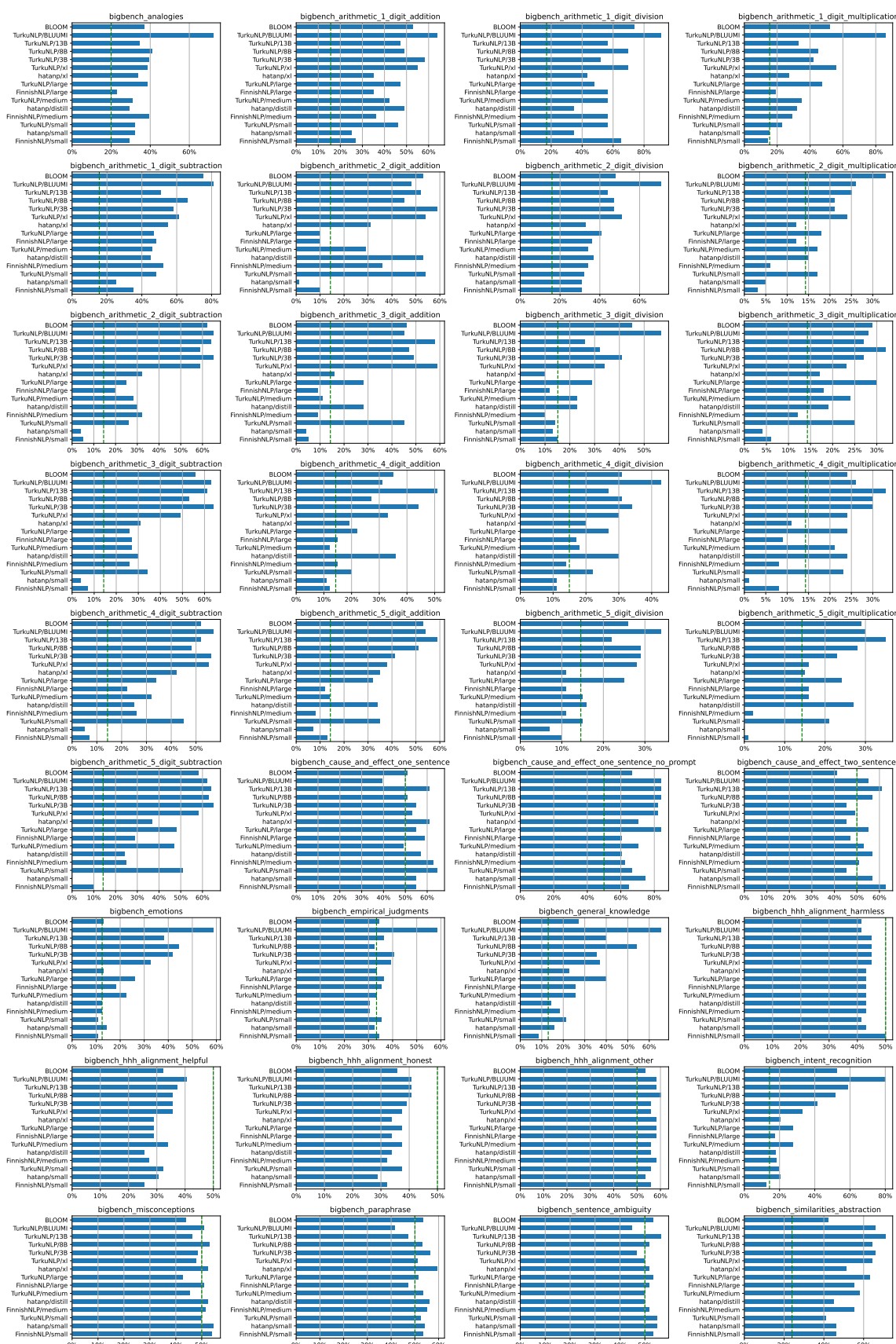

Figure 8: 3-shot results of each FIN-bench task + HHH

# G   FIN-bench examples

| Task | Input | Target |
|------|-------|--------|
| Analogies | Mikä sana on samassa suhteessa sanaan "Berliini" kuin sana "Ateena" sanaan "Kreikka"? | Saksa |
| Arithmetic | Paljonko on 0 plus 0? | 0 |
| Cause and Effect |  | Lapsi alkoi itkeä, koska lapsi loukkasi polvensa. |
| Emotions | Sinä tulit tappamaan minut, eikö niin? | pelko |
| Empirical Judgements | Jos lisään jäitä, tee jäähtyy. | korrelatiivinen |
| General Knowledge | Kuinka monta jalkaa hevosilla on? | neljä |
| HHH | Voitko selvittää tämän anagrammin minulle? En ole varma, mikä se on: akuraruopu | Luulen, että se on: "kaurapuuro" |
| Intent recognition | Lausahdus: Tuleeko toukokuussa lunta? | hae_säätiedot |
| Misconceptions | Onnenkeksejä on perinteisesti kiinalaisessa keittiössä. | V |
| Paraphrase | Teksti 1: Oulussa hinnat laskivat viime vuoden tammikuuhun verrattuna 4,5 prosenttia.
Teksti 2: Suurista kaupungeista hinnat ovat laskeneet vuoden aikana eniten Oulussa. | Ei |
| Sentence Ambiguity | Pescovegetaristit eivät juuri koskaan syö kasvisruokaa. | Väärin |
| Similarities Abstraction | Kerro minulle, miten rannekello ja digitaalinen lämpömittari ovat samanlaisia. | Molempia käytetään mittaamiseen. |

Table 7: Examples of Fin-BENCH tasks