# OpenReview forum: "FinGPT: Large Generative Models for a Small Language"
_EMNLP/2023/Conference — EMNLP 2023 Main_

### Official Review · Reviewer_yW8g · 2023-08-04

**Soundness:** 4

**Excitement:**

3: Ambivalent: It has merits (e.g., it reports state-of-the-art results, the idea is nice), but there are key weaknesses (e.g., it describes incremental work), and it can significantly benefit from another round of revision. However, I won't object to accepting it if my co-reviewers champion it.

**Paper Topic And Main Contributions:**

In this work, the authors build a large language model (LLM) for Finnish, a (relatively) low resource language. To this end, they gather data, preprocess it, and train a series of models based on the GPT architecture (186M-13B parameters) from scratch, as well as continuing the training of a BLOOM checkpoint (176B parameters). They also introduce FIN-bench, a derivative of BIG-bench for Finnish.

**Questions For The Authors:**

A) On lines 471-472, you say you compare two models and "find there is no significant difference in performance". Did you do any significance tests? If so, how did you do them?


**Reasons To Accept:**

While not introducing anything novel, I believe this work makes important contributions to the NLP community, namely:
* Well documented setup and recipe for building custom LLMs for smaller languages. The data sources used, the preprocessing and filtering steps are well documented, and the setup is well evaluated.
* Freely available checkpoints for the research community to build on. This is extremely important at a time where many models' weights (and details about training data) are not available.
* A variant of BIG-bench for Finnish.

**Reasons To Reject:**

Nothing major. As I said above, this work does not introduce anything novel nor does an analysis/study of prior models. Still, I found this paper a good starting point for anyone who wants to train LLMs.

**Reproducibility:**

4: Could mostly reproduce the results, but there may be some variation because of sample variance or minor variations in their interpretation of the protocol or method.

**Reviewer Confidence:**

3: Pretty sure, but there's a chance I missed something. Although I have a good feel for this area in general, I did not carefully check the paper's details, e.g., the math, experimental design, or novelty.

**Typos Grammar Style And Presentation Improvements:**

Line 53: remove "e.g.".
Line 129: you wrote "justText" instead of "jusText".
Table 4: "Mc4-Fi" -> "mC4-Fi".
Line 319: "GPUSs" -> "GPUs".
You included multiple appendices in the pdf that you do not refer to in the document. I understand the need to include more information to appease reviewers, but if you do not refer to them, they should be removed.

---

> ### Author Rebuttal · Authors · 2023-08-29
>
> Thank you for your careful review, positive comments, and helpful suggestions; we have addressed these as detailed below. We note that our perspectives differ on the novelty of our contributions and also present our view in this response.
>
> __Reasons To Reject:__ Nothing major. As I said above, this work does not introduce anything novel nor does an analysis/study of prior models. Still, I found this paper a good starting point for anyone who wants to train LLMs.
>
> __Response:__ Thank you for your feedback. We appreciate your recognition of the contributions of our work (“Reasons To Accept”) and find it unfortunate that you perceive the work as not introducing anything novel. While we believe we understand your perspective in observing that our work follows a previously established methodology with respect to model training, we would like to take this opportunity to note which parts of our work we consider novel contributions. First, specifically for Finnish, our contributions include creating the first large Finnish causal models (10B+ params), a substantial advance in state of the art for Finnish causal LMs, the first few-shot dataset for evaluating Finnish causal LMs, first quantitative analysis of the previously released “Finnish-NLP” and “hatanp” models on few-shot data, and introducing a novel pipeline, process, and preprocessed datasets for Finnish LLM training. More broadly, for low-resource languages in general, our work is one of several concurrent studies demonstrating that repeating data is a viable solution for data constraints for a low-resource language, while also observing some limitations of this approach. We also conducted an analysis of the efficacy of continued pretraining of BLOOM on a new language, providing additional support for the finding that very large language models can be successfully trained to adapt to learn a new language, which is in line with the preceding work on this topic.
>
> We acknowledge that our work does not introduce new methodology for pretraining, but believe that a study that increased the size of openly available models and advanced the state of the art for a language such as English or Chinese as much as ours does for Finnish would be generally considered meriting publication regardless of methodological contributions. While we understand that models for e.g. English are of greater interest to the EMNLP audience than ones for Finnish, we consider the access to models for smaller and lower-resourced languages critically important for advancing NLP for such languages and hope that EMNLP would also welcome work dedicated to smaller languages.
>
> Finally, we would like to briefly address the statement that our work does not include study of prior models. Of the models included in our evaluation, the “Finnish-NLP” and “hatanp” models are previously released dedicated models for Finnish, and BLOOM is a massively multilingual model created in prior work. Our work is to the best of our knowledge the first to assess either of the previously released Finnish causal models in few-shot settings, and we view our comparison of BLOOM performance to that of the BLUUMI variant that we create through continued pretraining to constitute comparison to a prior model. (We suspect there is some miscommunication on this point and would appreciate the opportunity to discuss this further e.g. to improve the clarity of the manuscript on the status of these models.)
>
>
> __Questions For The Authors:__
>
> __Question:__ On lines 471-472, you say you compare two models and "find there is no significant difference in performance". Did you do any significance tests? If so, how did you do them?
>
> __Response:__ Thank you for noting the ambiguity in our usage of “significant”. We performed the Shapiro-Wilkins test for normality (p=0.92 and 0.76) followed by Levene's test for equal variance (p=0.83) and then selected a suitable parametric test which in our case was an independent two-tailed t-test (p=0.92). Data used for comparison/testing was the model accuracy scores from eval-harness as shown in Figure 4. We now clarify that this statement reflects the result of a significance test in the manuscript.
>
> __Typos Grammar Style And Presentation Improvements:__
>
> __Comment:__ Line 53: remove "e.g.". Line 129: you wrote "justText" instead of "jusText". Table 4: "Mc4-Fi" -> "mC4-Fi". Line 319: "GPUSs" -> "GPUs". You included multiple appendices in the pdf that you do not refer to in the document. I understand the need to include more information to appease reviewers, but if you do not refer to them, they should be removed.
>
> __Response:__ Thank you for your careful reading! We have fixed the typos indicated here and will perform a further careful round of editing to assure none remain should the manuscript be accepted for publication. We consider all of the appendices to add meaningful information to our study and have now also corrected the oversight of not referencing them in the main body of the initially submitted version of the manuscript.

---

### Official Review · Reviewer_ddtQ · 2023-08-05

**Soundness:** 4

**Excitement:**

4: Strong: This paper deepens the understanding of some phenomenon or lowers the barriers to an existing research direction.

**Paper Topic And Main Contributions:**

This work creates a dataset for NLP studies of Finnish, as well as pertained transformer-based language models BLUMMI of different sizes.

**Questions For The Authors:**

- performance of smallest BLUMMI and FinnishNLP is higher than the 1/2/3-shot?
- Instead of the Finnish part of the popular multilingual pertaining corpora choices, the training data also includes Lonnrot which are more related to the out-of-copyright literature and Fiwiki which (may) include the timestamp of edition. It would be interesting to see if a temporal distribution/split of the corpora would be easily accessible since some of them are not as large as the entire multilingual resources? This question is also related to the bias measurement since the gender distribution of different professions in Finnish community might change over time.

**Reasons To Accept:**

- A high-quality compliment to the current English-based pertaining that paves the pathway for future Finnish research empower by pertained Finnish language models.
- Interesting experiments in diagnosing the alignment effects and potential social impacts like bias when applying the trained model in practice.

**Reasons To Reject:**

- Some examples of the evaluation benchmark can be added later to help draw a picture of the benchmark.

**Reproducibility:**

N/A: Doesn't apply, since the paper does not include empirical results.

**Reviewer Confidence:**

2: Willing to defend my evaluation, but it is fairly likely that I missed some details, didn't understand some central points, or can't be sure about the novelty of the work.

**Typos Grammar Style And Presentation Improvements:**

- The presentation of figure 3 and figure 5 can be improved.
- Since some of the original training corpora may not be explicitly relates to the Finnish culture (e.g. Fiwiki), while the bias measurement mainly compares with Finnish statistics, the results might be impacted by the Finnish corpora that are not directly relates to Finnish (according to Table 3, Fiwiki only takes a minor proportion, but it could be better to briefly address this).

---

> ### Author Rebuttal · Authors · 2023-08-29
>
> Thank you for your review, positive comments, and helpful suggestions for improving the manuscript.
>
> __Reasons To Reject:__ Some examples of the evaluation benchmark can be added later to help draw a picture of the benchmark.
>
> __Response:__  Thank you for the suggestion, we have added Appendix E that contains samples (input and target) from each task in FIN-bench. When addressing this suggestion, we additionally noted that we had mistakenly omitted the brief description of a single FIN-bench task from Section 5.1. This dataset (“Emotions”) consists of 160 samples and it was constructed based on an existing Finnish emotion dataset. The dataset was included in all evaluations as intended, and the erroneous omission of the description does not impact any of the conclusions of the manuscript. We apologize for the omission and thank the reviewer for prompting us to notice this oversight, which we have now corrected.
>
> __Questions For The Authors:__
>
> __Question:__ performance of smallest BLUMMI and FinnishNLP is higher than the 1/2/3-shot?
>
>
> __Response:__ Yes, the results indicate that for our smallest model and the FinnishNLP model, 0-shot performance is higher than the respective 1/2/3-shot performances. This could be related to a tendency of less capable models to simply copy patterns from preceding context, which in the case of few-shot examples can lead the models to copy whatever appears after “Answer:” (or equivalent) in the preceding examples. The larger models nevertheless demonstrate the expected pattern of being able to generalize from few-shot examples (i.e. 0-shot < 1-shot etc.), and we do not consider the limited capability of the smallest mode in this experiment problematic for the overall contribution of our work.
>
> __Question:__ Instead of the Finnish part of the popular multilingual pertaining corpora choices, the training data also includes Lonnrot which are more related to the out-of-copyright literature and Fiwiki which (may) include the timestamp of edition. It would be interesting to see if a temporal distribution/split of the corpora would be easily accessible since some of them are not as large as the entire multilingual resources? This question is also related to the bias measurement since the gender distribution of different professions in Finnish community might change over time.
>
> __Response:__ Absolutely! While we can unfortunately not reliably date more than a subset of the training material, we are currently preparing an analysis of the parts of the pretraining corpora for which accurate creation times can be attributed, and should our manuscript be accepted for presentation, we will include this additional perspective into the pretraining corpora.
>
> __Typos Grammar Style And Presentation Improvements:__
>
> __Comment:__ The presentation of figure 3 and figure 5 can be improved.
>
> __Response:__ Thank you for the observation. We have now revisited the layout and coloring of these figures, placing the legends clearly outside of the graphs, increasing font sizes when appropriate, and choosing a more distinctive color scheme. We have additionally replaced the original PNG figures with PDF-formatted figures in the manuscript to assure that the figures remain clear when zoomed in.
>
> __Comment:__ Since some of the original training corpora may not be explicitly relates to the Finnish culture (e.g. Fiwiki), while the bias measurement mainly compares with Finnish statistics, the results might be impacted by the Finnish corpora that are not directly relates to Finnish (according to Table 3, Fiwiki only takes a minor proportion, but it could be better to briefly address this).
>
> __Response:__ While there may be articles in the Fiwiki corpus that have been at least initially translated from other languages, the Finnish Wikipedia from which this corpus is derived is primarily created by Finnish volunteers and, in our experience with the resource, broadly reflects Finnish culture and interests. We would thus not anticipate the inclusion of texts drawn from the Finnish Wikipedia to bias the model to be less representative of Finnish culture. (We are not entirely sure we have understood this comment correctly and are happy to discuss and revise accordingly if not.)

---

### Official Review · Reviewer_Y78r · 2023-08-05

**Soundness:** 4

**Excitement:**

4: Strong: This paper deepens the understanding of some phenomenon or lowers the barriers to an existing research direction.

**Missing References:**

- Line 300 - missing citation for FinCORE
- Line 479-480 - missing citation for Big-Bench HHH


**Paper Topic And Main Contributions:**

Training large language models (LLMs) for NLP tasks has become ubiquitous. Given the majority of the research concentration in high resource languages, mainly English, the current work aims to bridge this gap by building LLMs of varying sizes for Finnish. They create the large monolingual corpora, train LLMs of varying sizes ranging from 186M to 13B, extend the BLOOM model to Finnish and also create Fin-Bench, a derivative from Bing-Bench. While the work is for a single language, the ideologies can be extended to other languages.

**Questions For The Authors:**

- Line 222-224 - Do you mean to say you have individually performed deduplication on each of the individual subsets or deduplication after merging all the sources. If it’s the former, what is the overlap between mC4 and CC-Fi? And has this been removed?
- Line 227-229 - Wouldn’t trimming the beginning and ending of the document disturb the flow of the document?
- Line 250-251 - if this a typo or is the threshold for perplexity actually 100k?
- Lines 281-285 - 131,072 looks to be a large vocabulary for a single language. While I understand the agglutinative nature of the language, were other sizes of vocabulary also tested?
- Lines 377-380 - What is the process of correcting the translations and checking cultural relevance. Is this a human process? What are the guidelines provided
- Lines 428-431 - Was this a manual process?
- This is not a major concern but have the authors tried translate-test with English-only models? That is, translate the Finnish to English and perform inference with English-only models. This could be a strong baseline to motivate the need to build a Finnish only LLM
- Have you tried to compare zero-shot testing of fin-bench with popular English LLMs? (the idea being is it enough to translate-test or do we need a language specific LLM)
- Information missing on how the test sets were created. Lines 377-380 mention that they are machine translated, while lines 393-396 says they are manually translated. There needs to be clear explanations of how the test sets were created and checked for any mistakes
- Not important in the context of acceptance but what does BLUUMI stand for?

**Reasons To Accept:**

- The authors have done the full cycle of collecting data, cleaning the data, cleaning for PIIs, trained models of varying sizes and also released all the scripts for the processes.
- While they created standalone Finnish models, they also extended the BLOOM model to add Finnish and observed no degradation in English performance
- Beyond the task level testing the authors also test for biases, human alignment and toxicity generated by the model, which gives a hint of how and where to use them and cautions at using in production systems.


**Reasons To Reject:**

- Testsets sizes are very small with only 2 tasks having more than 200 examples. While this is not entirely bad, it would’ve helped to anaylze the patterns in the generations taking advantage of the smaller testset sizes. Since the test set sizes are small a qualitative analysis would be better than quantitative analysis.


**Reproducibility:**

3: Could reproduce the results with some difficulty. The settings of parameters are underspecified or subjectively determined; the training/evaluation data are not widely available.

**Reviewer Confidence:**

4: Quite sure. I tried to check the important points carefully. It's unlikely, though conceivable, that I missed something that should affect my ratings.

**Typos Grammar Style And Presentation Improvements:**

Typically “Pretraining” section is used to provide motivation for the choice of hyperparamters, architecture choices and hardware. I feel that lines 311-349 can be condensed to have the important aspects of hardware and rest moved to appendix. And use this space to expand more on results and observations
- But I would definitely commend the authors for adding these finer details into the paper

I would suggest reformatting the tables, while there is absolutely no problem with the current tables, having top and bottom margin makes it look much cleaner. And try to maintain uniform font sizes inside the tables.

---

> ### Author Rebuttal · Authors · 2023-08-29
>
> Thank you for your careful review and the many valuable points you raise. We have addressed these to the best our our ability as detailed below.
>
> __Reasons To Reject__: Testsets sizes are very small with only 2 tasks having more than 200 examples. While this is not entirely bad, it would’ve helped to analyze the patterns in the generations taking advantage of the smaller testset sizes. Since the test set sizes are small a qualitative analysis would be better than quantitative analysis.
>
> __Response__: We agree that the comparatively small sizes of individual tasks in FIN-bench limit the ability to draw quantitative conclusions when the tasks are considered in isolation. This is in part why we have chosen to focus only on the aggregate performance across all FIN-bench tasks in the main body of the manuscript (Section 5.2) and avoid any claims drawn only from results on individual tasks (included in Appendix D for reference). We have now updated the description of FIN-bench to also include the total number of examples in the dataset, 3919. We note that the comparatively small sizes of individual tasks represents a limitation inherited directly from the original BIG-Bench, as our dataset consists primarily of translations from the original and was created with the goal of keeping these individual tasks similar to the originals.
>
> __Questions For The Authors:__
>
> __Question:__ Line 222-224 - Do you mean to say you have individually performed deduplication on each of the individual subsets or deduplication after merging all the sources. If it’s the former, what is the overlap between mC4 and CC-Fi? And has this been removed?
>
> __Response:__ The former, i.e. the sources have only been deduplicated separately. Due to the size of the datasets and differences in the HTML-to-text extraction processes, postprocessing, and metadata in mC4 and CC-Fi, we were regrettably not able to produce a reliable estimate of the true overlap in text content between these two sources in the time allotted for this response. Should our manuscript be accepted for publication, we will include information on this overlap as well as details of the estimation process in the camera-ready version of the manuscript.
>
> __Question:__ Line 227-229 - Wouldn’t trimming the beginning and ending of the document disturb the flow of the document?
>
> __Response:__ Yes, this will certainly happen in some cases. Deduplication approaches that include the option of keeping only part of a document inevitably risk some damage to the flow of the primary text content. Our choice of approach here aims to minimize such damage by assuring that any remaining text was continuous in the source document while still eliminating highly redundant sections. The choice also reflects the observation that the beginnings and ends of documents drawn from web sources frequently include boilerplate material appearing identically over many pages (menus, site descriptions, etc.).
>
> __Question:__ Line 250-251 - if this a typo or is the threshold for perplexity actually 100k?
>
> __Response:__ The value 100K is correct. Finnish is a morphologically rich agglutinative language, leading to a comparatively high number of unique word forms, which in turn cause word-level n-gram models such as that applied here to give comparatively high perplexity values.
>
> __Question:__ Lines 281-285 - 131,072 looks to be a large vocabulary for a single language. While I understand the agglutinative nature of the language, were other sizes of vocabulary also tested?
>
> __Response:__ Our choice of a comparatively large vocabulary was primarily informed by our previous work on language models for Finnish. We additionally performed an informal study comparing the vocabulary items added when increasing the vocabulary to this size from 64K to assure that the new items are (mostly) valid Finnish word forms.
>
> __Question:__ Lines 377-380 - What is the process of correcting the translations and checking cultural relevance. Is this a human process? What are the guidelines provided
>
> __Response:__ Yes, this was a fully manual process carried out by native Finnish speakers. Annotators were given informal guidelines about correcting translations and cultural relevance.
> For the former the general guideline was to maintain the original style and meaning as closely as possible and correct any errors arising from the machine translation process.
> For the latter annotators were asked to change US and English-centric examples into Finnish-centric ones to maintain e.g. the level of general knowledge required to perform the task. This was found out to be fairly subjective, and many individual cases were examined by multiple annotators to land on a collective agreement.
> As a simple example of a change made to the content, the (false) claim "The capital of the United States is and always has been Washington, D.C'' (sentence_ambiguity task) was revised to (the equally false) "The capital of Finland is and always has been Helsinki" (backtranslated from Finnish).
>
>
> __Question:__ Lines 428-431 - Was this a manual process?
>
> __Response:__ Yes. this was done manually
>
>
> __We respond to the following two related questions together.__
>
> __Question:__ This is not a major concern but have the authors tried translate-test with English-only models? That is, translate the Finnish to English and perform inference with English-only models. This could be a strong baseline to motivate the need to build a Finnish only LLM
>
> __Question:__ Have you tried to compare zero-shot testing of fin-bench with popular English LLMs? (the idea being is it enough to translate-test or do we need a language specific LLM)
>
> __Response:__ This is an excellent suggestion, and while we regularly work with machine translation (e.g. in creating FIN-bench in this study), we have not been able to perform a formal comparison as part of this study due to time and space constraints. We hope to be able to address this suggestion in future work. Generally speaking, it is our impression that state-of-the-art machine translation systems for Finnish introduce errors that impact the understandability of these types of tasks with non-trivial frequency. We would also like to note that in our informal assessment the fluency of the Finnish generated by our models surpasses that of automatic translations from English, providing another motivation for using these models apart from the ability to perform zero- and few-shot tasks with Finnish input.
>
> __Question:__ Information missing on how the test sets were created. Lines 377-380 mention that they are machine translated, while lines 393-396 says they are manually translated. There needs to be clear explanations of how the test sets were created and checked for any mistakes
>
> __Response:__ Thank you for the comment. In the submitted version lines 374-379 meant to separate these tasks from the machine translated ones (the majority of the tasks). We have now further clarified this point in the manuscript.
>
> __Question:__ Not important in the context of acceptance but what does BLUUMI stand for?
>
> __Response:__ BLUUMI can be read either as a finnicized version of BLOOM or as a combination of the names BLOOM and LUMI (the supercomputer used to create the models).
>
> __Missing References:__
>
> __Comment:__ Line 300 - missing citation for FinCORE
>
> __Response:__ Thank you for noting this. The model and the FinCORE corpus referenced on this line were both introduced in the cited manuscript (Skantsi and Laippala, 2022), but the text failed to make this clear. We have now revised the relevant text to clarify this point.
>
> __Comment:__ Line 479-480 - missing citation for Big-Bench HHH
>
> __Response:__ Thank you for noting this, we have now added this missing citation.
>
> __Typos Grammar Style And Presentation Improvements:__
>
> __Comment:__ Typically “Pretraining” section is used to provide motivation for the choice of hyperparamters, architecture choices and hardware. I feel that lines 311-349 can be condensed to have the important aspects of hardware and rest moved to appendix. And use this space to expand more on results and observations
> But I would definitely commend the authors for adding these finer details into the paper
>
> __Response:__ Thank you for the suggestion! Should the manuscript be accepted for publication, we will be able to use an additional page to react to reviewer comments, which we estimate should allow us to keep the current details in the main body of the manuscript while also adding the further requested information. If we are unable to fit all of the additional material to the main manuscript, we will follow your helpful suggestion on how to reorganize the material to save space.
>
> __Comment:__ I would suggest reformatting the tables, while there is absolutely no problem with the current tables, having top and bottom margin makes it look much cleaner. And try to maintain uniform font sizes inside the tables.
>
> __Response:__ Thank you, we have reformatted all tables to have margins and uniform font sizes, as suggested.

---

### Meta-Review · Area_Chair_bx4n · 2023-09-29

**Recommendation:** 5

**Metareview:**

Solid contribution to the open-source large language models for low-resourced languages -- Finish in this case. Extensive analysis, translated evaluation set and great results demonstrating how LLMs could be used effectively for low-resource languages.

---

### Decision · Program_Chairs · 2023-10-07

**Decision:**

Accept-Main

**Comment:**

Solid contribution to the open-source large language models for low-resourced languages -- Finish in this case. Extensive analysis, translated evaluation set and great results demonstrating how LLMs could be used effectively for low-resource languages.